# Sub-THz Vibrational Dynamics in Ordered Mesoporous Silica Nanoparticles

**DOI:** 10.3390/nano13142078

**Published:** 2023-07-15

**Authors:** Eduardo Hernando Abad, Frédéric Bouyer, Laroussi Chaabane, Alan Zerrouki, Jérémie Margueritat, Lucien Saviot

**Affiliations:** 1Laboratoire Interdisciplinaire Carnot de Bourgogne, UMR 6303 CNRS–Université de Bourgogne Franche Comté, 21000 Dijon, France; 2Institut Lumière Matière, UMR5306, Université de Lyon, 69622 Villeurbanne, France

**Keywords:** mesoporous silica, inelastic light scattering, phononic crystal

## Abstract

The vibrational dynamics in the sub-THz range of mesoporous silica nanoparticles (MSNs) having ordered cylindrical mesopores was investigated. MCM-41 and SBA-15 particles were synthesized, and their structure was determined using scanning electron microscopy (SEM), low-angle X-ray diffraction (XRD), N_2_ physisorption analyses, and Raman scattering. Brillouin scattering measurements are reported and enabled determining the stiffness of the silica walls (speed of sound) using finite element calculations for the ordered mesoporous structure. The relevance of this approach is discussed based on the comparison between the numerical and experimental results and previous works reported in the literature.

## 1. Introduction

Wave propagation in periodic structures has attracted much attention during the last few decades. In the case of elastic or acoustic waves, such structures are called phononic crystals [1,2]. Their frequency of operation increases as their spacial periodicity decreases. Current fabrication methods make it possible to design phononic crystals in the GHz range. Higher frequencies require spacial periodicity in the few nanometers range, which is difficult to attain with commonly used top-down processes such as lithography. Bottom-up methods can provide alternate routes to obtain such periodic structures. The chemical synthesis of ordered mesoporous oxides such as silica and titania is interesting in this context. Their periodic structure results from the periodic arrangement of the micelles formed by the surfactant precursors. Silica or titania condensation occurs around the micelle template. In particular, mesoporous silica has been investigated extensively. Different lattice structures, spacial periodicities, and pore diameters can be obtained depending on the synthesis conditions and the precursors. The mesopores can also be filled by a different material. As a result, a large variety of ordered mesoporous structures are in principle possible. Mesoporous silica is most often synthesized as nanoparticles (MSNs) having a diameter much larger than the periodicity of the mesopores. For example, MSNs with diameters in the 100 nm range have attracted much attention for their potential applications in drug delivery [3]. X-ray diffraction (XRD) patterns and transmission electron microscopy (TEM) images confirm their high-quality periodic structures extending over hundreds of nanometers or more. Oriented mesoporous silica films have also been prepared [4]. These may be more attractive for phononic applications.

The vibrational dynamics of mesoporous materials has been studied previously [5,6,7,8]. However, to the best of our knowledge, the periodicity of the mesoporous structure has never been explicitly taken into account. In a recent work [9], some of us investigated mesoporous titania using Raman scattering including in the low-wavenumber range down to $∼10{\mathrm{cm}}^{-1}$ ( 300 GHz). Two peaks were observed in the sub-THz range (∼0.4 and 0.7 THz). They were tentatively assigned to vibrations of free titania nanoparticles and nanowires, which were identified in the mesoporous structure. However, the Raman peaks due to optical phonons were reminiscent of those of bulk anatase TiO_2_ without the significant shifts and broadening expected for anatase nano-objects [10]. Indeed, mesoporous titania is not an ensemble of nano-objects. The TEM images were consistent with a disordered set of cylindrical holes going through bulk anatase. The validity of the simple approach based on isolated nano-objects is, therefore, questionable due to the probable strong coupling between them in the mesoporous structure.

The goal of the present study was to propose a method to take into account explicitly the ordered mesoporous structure. To do so, we turned to mesoporous silica, whose ordered mesoporous structure is better known. We considered the MCM-41 structure, which consists of a 2D hexagonal lattice of cylindrical pores. A typical MCM-41 MSN and a cross-section of the hexagonal lattice of pores are schematized in Figure 1. We synthesized MCM-41 MSNs, calcined them at different temperatures, and characterized their mesoporous lattice structure (XRD and N_2_ physisorption), their shape and size (SEM), and the composition and structure of the silica walls (Raman scattering). Acoustic vibrations were probed by Brillouin scattering. We interpreted the obtained spectra with calculations based on a phononic crystal approach. This method relies on a continuum isotropic elasticity description of bulk silica to calculate the phonons associated with the ordered mesoporous structure using the finite element method [11]. This comparison makes it possible in principle to determine the elastic parameters inside the silica walls. However, these parameters are not well-known in such thin walls, making it difficult to confirm the validity of our approach. For this reason, we considered MSNs calcined at rather high temperatures so that the elastic parameters should tend toward those of bulk silica. At the same time, the range of calcination temperatures was selected to avoid any damage to the mesoporous structure. For comparison purposes, we also investigated SBA-15 MSNs, presenting a similar 2D hexagonal lattice of cylindrical pores with a larger spatial periodicity and non-porous silica nanoparticles (NPs).

## 2. Materials and Methods

### 2.1. Materials

Cetyltrimethylammonium bromide (CTAB, 99%), Pluronic P123 (EO_20_PO_70_EO_20_, Mn = 5800 g/mol), a 2 M NH_3_ ammonia solution in ethanol, and hydrochloric acid (37 wt% and 1 M standard solution) were obtained from Sigma-Aldrich (St. Louis, MO, USA). A 2 M NaOH carbonate-free standard solution, absolute ethanol (99%), and tetraethoxysilane (TEOS, 99%) were supplied by Thermo Fisher Scientific (Weltham, MA, USA). All chemicals were used as received.

### 2.2. Synthesis of Porous and Non-Porous Silica

MCM-41 mesoporous silica nanoparticles (MCM-41 MSNs) were prepared according to Varache et al. [12] with slight modifications. Briefly, 416.6 mg of CTAB ( 1.14 mmol) was dissolved in 200 mL of ultrapure water ( 18 MΩ · cm, Elga (High Wycombe, UK)) in a 500 mL three-neck round-bottom flask. The solution was purged thrice with a N_2_ flow ( 1 L /min), and 1.45 mL of 2 M NaOH standard solution protected from ambient air ( 2.90 mmol) was added under the same N_2_ flow. After stirring at 80 °C for 1 h under a N_2_ flow of 150 mL/min, 2.01 mL of TEOS ( 9 mmol) was added at 1 mL/min under a N_2_ flow of 1 L/min. The molar composition of the solution was TEOS:NaOH:CTAB:H_2_O = 1:0.32:0.127:1235. After 2 h of stirring, the pH of the colloidal suspension was adjusted to ∼7 by adding 3 mL of 1 M HCl solution. The mixture was stirred for another 24 h under a static N_2_ atmosphere. The sample was separated by centrifugation (10,000 *g*, 1 h). The solid cake was redispersed in ultrapure water using a sonicator, and the centrifugation–redispersion procedure was repeated once before removing water by rotary evaporation. Then, 100 mg of powder was calcined under static air at 550 °C, 700 °C and 800 °C for 6 h with a heating rate of 1 °C/min.

SBA-15 mesoporous silica powder was synthesized according to Yuan et al. [13] with minor modifications. 3 g of Pluronic P123 ( 0.52 mmol) was dissolved in 80 mL of HCl 2 M for 2 h at 40 °C. Then, 6.7 mL of TEOS ( 30 mmol) was added at once. The mixture was stirred for 24 h at 40 °C and aged for another 24 h at 100 °C in a glass bottle. The molar composition of the solution was TEOS:HCl:P123:H_2_O = 1:5.3:0.017:148. The white solid was then filtered and washed several times using ultrapure water until the conductivity of the supernatant was close to that of water. After drying, 100 mg of powder was calcined using the same conditions as for the MCM-41 MSNs.

Monodisperse non-porous silica nanoparticles were synthesized using the Stöber procedure [14]. At room temperature, 44.6 mL of TEOS was added dropwise in a solution of 725.4 mL of absolute ethanol, 90 mL of water, and 140 mL of 2 M ammonia in ethanol. The molar composition was TEOS:H_2_O:NH_3_:EtOH = 1:25:1.4:62. The mixture was stirred magnetically for 24 h at 500 rpm. After centrifugation, the particles were dried for one night.

### 2.3. Characterization of the Silica Nanomaterials

The textural properties of the samples were characterized by nitrogen physisorption analysis at 77 K using a Micromeritics ASAP 2020 setup (Norcross, GA, USA). The specific surface area of the materials was calculated using the BET method [15] in the relative pressure range between 0.06 and 0.25. The pore-size distribution was determined from the desorption branches of the isotherms using the BJH method [16] at $P/P_{0}\leq 0.75$. Powder XRD patterns of the SBA-15 and MCM-41 samples were collected using a Bruker D8 Advance diffractometer (Billerica, MA, USA) equipped with Vantec linear PSD, using CuKα radiation (35 kV, 35 mA). The porous organization of the samples was analyzed between 2θ=0.5 and 10°, with a 0.017° step and a step time of 3 s. The particles were characterized by SEM (Hitachi (Tokyo, Japan) SU8230) at 10 kV and a distance of 11.7 mm. Raman spectra were measured in the back-scattering geometry using a Renishaw inVia setup (Wotton-under-Edge, UK) with a 532 nm laser and a ×50 microscope objective (numerical aperture NA = 0.75). Brillouin spectra were measured using a tandem Fabry–Perot from JRS Scientific Instrument (Zwillikon, CH) in the back-scattering geometry with an inverted microscope, a 660nm laser, and a ×100-long working distance microscope objective (NA = 0.9).

## 3. Results

### 3.1. MCM-41 MSNs

The shape and size of the calcined MCM-41 MSNs were determined from SEM images. A representative image of the sample calcined at 700 °C is shown in Figure 2 (right). The NPs are approximately spherical, even if some facets can be seen. Their average diameter $\bar{d}$ was determined from the size distributions (Figure 2 left) and reported in Table 1. It decreased with the increasing calcination temperature. We assigned this result to the incomplete condensation of silica after synthesis, which occurred during calcination.

The XRD patterns in Figure 3 correspond to the expected hexagonal structure of MCM-41 mesoporous silica. The hk indices of the peaks were 10, 11, 20, and 21 with increasing angles [17]. Their observation up to 700 °C attested to the highly regular ordered structure. At 800 °C, the intensity of the 11 and 20 peaks was significantly smaller and the 21 peak was absent, indicating a less-ordered structure. The shift of the peak positions towards larger 2θ values with increasing temperature indicated that the center-to-center pore spacing *a* calculated according to $a=d_{10}\frac{2}{\sqrt{3}}$ decreased. This decrease agreed with the shrinkage of the MSNs observed with SEM. The values are reported in Table 1.

The pore diameters ($d_{p}$) were determined from the N_2_ physisorption isotherms (Figure 3, right), which were type IV, characteristic of MCM-41 materials. The pore size (see Table 1) decreased significantly with increasing calcination temperature. The wall thickness $e=a-d_{p}$ did not change significantly when varying the temperature of calcination.

The different parameters describing the lattice structure of MCM-41 MSNs are presented in Table 1. Porosity was calculated from the lattice parameter and the pore diameter using Equation (Equation 1). Note that the structure was ill-defined when the cylindrical pores overlapped, i.e., when $d_{p}\geq a$ (see Figure 1). Therefore, $0\leq p<\frac{\pi }{2\sqrt{3}}≃0.9$.
(1)$$p=\frac{\pi }{2\sqrt{3}}{\left(\frac{d_{p}}{a}\right)}^{2}$$

The samples were also characterized using Raman scattering (see Figure 4). The spectrum of the sample before calcination showed mostly features related to the chemical moieties used to form the mesopores, most notably the C–H stretching vibrations near 2900 cm^−1^, a photoluminescence background, and a weak contribution from amorphous silica. After calcination, only the Raman peaks related to amorphous silica remained [18,19,20,21,22]. The spectra were normalized on the SiO_2_ network peaks (broad features near 440 and 800 cm^−1^). The resulting intensity of the D_1_ peak at 492 cm^−1^ did not change significantly when increasing the calcination temperature, while the one of the D_2_ peak at 605 cm^−1^ increased and the one of the SiOH band near 980 cm^−1^ decreased. This showed that the OH groups were eliminated even above 700 °C. The increase of the intensity of the D_2_ band, attributed to the symmetric stretch breathing mode of the O atoms in a threefold planar ring of Si–O bonds, was assigned to an increase of the number of such rings. The fourfold planar rings at the origin of the D_1_ band were not affected by calcination. Note also that, for clarity, a constant background was removed from all the spectra in Figure 4. This photoluminescence background in the calcinated samples may come from oxygen-related defects (non-bridging oxygen hole centers) as in non-porous amorphous silica [23].

Another way to check for the structure of silica consists of calculating $V_{{\mathrm{SiO}}_{2}}$, the volume of the SiO_2_ walls inside an average NP at each calcination temperature. The weight loss being small above 550 °C, variations in $V_{{\mathrm{SiO}}_{2}}$ after calcination corresponded to variations of the mass density due to a modification of the structure of silica. Using Equation (Equation 2), we obtained $V_{{\mathrm{SiO}}_{2}}≃460$, 418, and 322 × 10^3^ nm^3^ at T=550, 700, and 800 °C, respectively. This volume variation showed that the structure of the walls was affected by the calcination temperature. Raman spectroscopy indicated that this was related to the elimination of hydroxyl groups and the rearrangement of the silica network, probably close to the surface of the mesopores (creation of threefold rings). We expected the mass of the MSN NPs to first decrease with the calcination temperature and then to remain constant. The significant decrease of $V_{{\mathrm{SiO}}_{2}}$ between 700 and 800 °C may indicate that the ordered structure was affected as well, as already pointed out for the XRD and BET measurements.
(2)$$V_{{\mathrm{SiO}}_{2}}=\frac{\pi {\bar{d}}^{3}}{6}\left(1-p\right)$$

### 3.2. SBA-15 MSNs and Non-Porous Silica NPs

Using a similar procedure, we determined *a* and $d_{p}$ from the XRD patterns and the N_2_ physisorption isotherms of the SBA-15 samples. The results are summarized in Table 2. The corresponding experimental data are presented as the Appendix A. The results obtained after calcination at 800 °C did not differ significantly from those at 550 and 700 °C, indicating that the ordered mesoporous structure was preserved, contrary to the MCM-41 NPs. The Raman spectra showed the same tendency as the ones of the MCM-41 NPs, namely that the organic moieties were removed after calcination and that the D_2_ peak intensity increased with the calcination temperature. The SEM images showed that the size and shape distributions were quite large. Most of the NPs were elongated with a diameter of about 0.5 μm and lengths of a few μm.

We also performed SEM and Raman measurements on the non-porous SiO_2_ NPs. The corresponding experimental data are presented as the Appendix A. In this case, the NPs were almost spherical with a narrow size distribution ($\bar{d}=131\mathrm{nm}$). The Raman peaks showed the presence of amorphous silica. Some organic moieties were still present due to the absence of calcination. The condensation of silica was also less complete, as attested by the relative intensity of the Si–OH band.

### 3.3. Brillouin Scattering

The Brillouin spectra of the samples are presented in Figure 5. Peaks are clearly visible in all cases, even if they appear only as shoulders for the larger SBA-15 MSNs. The spectrum of non-calcined MCM-41 MSNs presented a shoulder at a very low frequency. In this case, we cannot rule out a contribution from the organic moieties inside the pores. Since we were interested in the vibrational dynamics of the silica structure in this work, we will not consider this spectrum in the following.

The Brillouin peak frequencies were determined by fitting the spectra using a Lorentzian line shape. Details regarding the fitting procedure are given as the Appendix A. The obtained peak frequencies are reported in Table 1 and Table 2 for the MCM-41 and SBA-15 MSNs. For the non-porous SiO_2_ NPs, the Brillouin peak’s maximum was at 15 GHz.

### 3.4. Calculation of the Phonon Bands

To interpret the Brillouin results, we calculated the phonons branches for the hexagonal structure depicted in Figure 1 (right) using the numerical approach described by Laude [11] with FreeFEM 4.11 [24]. The goal was to obtain the frequencies of the phonons in the ordered mesoporous structure, which were required to interpret the Brillouin peaks.

Amorphous silica was modeled using an isotropic continuum elasticity approximation. The validity of such an approximation inside silica walls a few nanometers thick (see Table 1 and Table 2) was of course questionable, in particular because of the large surface/volume ratio of the mesoporous structures. However, we note that continuum elasticity had been shown to be suitable to model the lowest-frequency vibrations of smaller nanocrystals [25] and thinner nanoplatelets [26]. In addition, even if this approximation failed to provide accurate quantitative predictions, we believe that it would still be relevant to qualitatively understand the role of the main parameters, i.e., the variations with the geometrical parameters (*a*, $d_{p}$) and the elastic constants of the silica walls.

Two parameters are needed to describe the isotropic continuum elasticity of bulk amorphous silica. The Brillouin peak frequencies are generally proportional to the sound speed in the medium. For this reason, the longitudinal ($c_{L}$) and transverse ($c_{T}$) sound speeds were used in this work. In addition, this choice factored out the mass density of the silica walls in the following. The peak in the Brillouin spectrum of our SiO_2_ NPs in Figure 5 was at 15 GHz. Using the sound speeds obtained in a previous work for similar NPs [27] ($c_{L}=4220$ and $c_{T}=1420\;m/s$), we calculated the frequency of the spheroidal mode with ℓ=2 for a 131 nm nanosphere at 17.6 GHz. The observed frequency was lower, which we assigned to the different synthesis conditions of the NPs and, in particular, to the incomplete condensation of silica discussed before, which resulted in a smaller stiffness and smaller sound velocities. Different sound speeds for amorphous silica are reported in the literature depending on the preparation conditions. We note that, in Still et al. [27], $\frac{c_{L}}{c_{T}}≃1.52$ (corresponding to Poisson’s ratio ≃0.12), which is close to the values reported for fused silica [28,29]. We assumed that this ratio was preserved in all our samples. In order to account for the variability of $c_{L}$, we present the calculated phonons’ frequencies in reduced units whenever possible. Then, $c_{L}$ was determined from the experimental results to match the observed frequencies. We first applied this procedure to the non-porous silica NPs. Their experimental Brillouin frequency ( 15 GHz) was reproduced with $c_{L}≃3600\;m/s$.

Figure 6 presents the phonon bands calculated for porosity p=0.35. The acoustic and optical branches of the phononic crystal were observed, having a zero or non-zero frequency at Γ, respectively. The slope of the acoustic branches at Γ was proportional to the sound speed in the mesoporous structure for waves propagating along the corresponding direction. For comparison, two lines are plotted corresponding to $c_{L}$ and $c_{T}$ in bulk amorphous silica. The sound speeds were smaller in the porous structure even if the difference was quite small for phonons propagating in the direction of the cylindrical pores (*z*). Multiple optical phonon branches can be seen. To highlight breathing-like vibrations, the variation during the vibration of the silica surface area in the hexagonal cell was calculated and plotted using variable-size symbols (blue) [30]. Breathing-like bands are clearly visible near $\nu ∼\frac{1.3c_{L}}{a}$.

#### 3.4.1. Acoustic Branches

From the slope of the three acoustic branches, we calculated the sound speeds and derived all the corresponding hexagonal elastic coefficients ($C_{ij}$). Note that, for hexagonal elasticity, the sound speeds depend on the angle between *z* and the propagation direction only. The obtained variations of the sound speeds as a function of porosity are shown in Figure 7 for two propagation directions. The longitudinal sound speed of waves propagating along *z* were almost independent of porosity. The speeds of transverse waves propagating along *z* decreased slightly. Waves propagating in the xy plane were much more affected by porosity.

The elastic parameters obtained from the slope of the acoustic branches near Γ provided an elastic approximation suitable for long wavelength acoustic phonons. We used this approximation to calculate the eigenfrequencies of an elastic sphere, which was large compared to the mesoporous periodicity ($\bar{d}\gg a$). We were interested in the spheroidal vibrations with angular momentum ℓ=2 corresponding to the main features [31] observed in the Brillouin measurements (Figure 5). These ℓ=2 eigenmodes (degeneracy 2ℓ+1=5) split into A_1g_+E_1g_+E_2g_ due to the hexagonal symmetry (point group D_∞h_), corresponding to the m=0, m=±1, and m=±2 modes, respectively. The frequency of these modes was calculated using the Rayleigh–Ritz variational method [30,32], and their variation with porosity is presented in Figure 8. The three frequencies decreased significantly with increasing porosity. The E_1g_ vibrations were the most-affected by porosity because their deformation occurred in the xy plane, contrary to the A_1g_ and E_2g_ ones.

#### 3.4.2. Optical Branches

In crystals, optical phonons near the center of the Brillouin zone can give rise to Raman scattering. Similarly, the optical branches in Figure 6 may result in additional inelastic light scattering peaks at frequencies higher than those of the acoustic phonons discussed before, but still in the sub-THz range. To further investigate this possibility, we determined the irreducible representations of the vibrations at the center of the Brillouin zone (Γ, point group D_6h_) and focused on the first few Raman active ones. To highlight the motion associated with each of these modes, we plotted the displacement of the surface of the pores in Table 3 for p=0.35. The first Raman active mode (E_2g_) had a quadrupolar-like deformation of the surface of the pore (stretching along one direction). The first totally symmetric one (A_1g_) was breathing-like (stretching along all the directions in the xy plane). The other modes had a more-complex nature.

We also checked the dependence of the optical phonons on porosity. The frequency variations are shown in Figure 9. The changes with porosity for all the branches were quite complex, but we note that the frequency of the quadrupolar-like mode depended only weakly on porosity. In addition, the reduced frequency of the breathing-like mode varied between 1.0 and 1.9. This provided relatively narrow frequency ranges where the signatures of these optical phonons may be found.

## 4. Discussion

For the MCM-41 MSNs, we deduced the sound speed inside the silica walls by assuming that the frequency at the maximum of the observed Brillouin peak frequency corresponded to the average of the quadrupolar-like eigenfrequencies (Figure 8). We obtained $c_{L}≃3110$, 3510, and 3000 m/s for T=550, 700, and 800 °C. The stiffness of the silica walls increased with the calcination temperature from 550 to 700 °C, as expected. We assigned the anomalous decrease of $c_{L}$ at 800 °C to the less-ordered mesoporous structure as discussed before, which can lead to significant deviations from the perfect phononic crystal model. $c_{L}=3510\;m/s$ obtained at 700 °C was close to the reported value for non-porous SiO_2_ NPs [27] and even closer to $c_{L}=3600\;m/s$ obtained above for our non-porous SiO_2_ NPs. This rather good agreement supported the phononic crystal approach.

For the SBA-15 MSNs, interpreting the Brillouin peaks as acoustic phonons confined in the MSN NPs was problematic. Indeed, elastic waves confined along the shortest length of the elongated NPs (about 500 nm) had a much lower frequency than the observed peaks. In the following, we assumed that the NPs were large enough to neglect this confinement. The Brillouin frequency is, therefore, expressed, as in the bulk material, as:(3)$$\nu_{B}=\frac{2cn_{\mathrm{eff}}}{\lambda }$$
where λ is the optical wavelength, $n_{\mathrm{eff}}$ the effective refractive index, and *c* the longitudinal sound speed in the mesoporous material. Both $n_{\mathrm{eff}}$ and *c* varied with porosity *p*. The effective optical index was obtained from the Maxwell–Garnett model for a bi-component medium (*n*–air):(4)$$n_{\mathrm{eff}}\left(p\right)=n\sqrt{\frac{1+2n^{2}+2p\left(1-n^{2}\right)}{1+2n^{2}-p\left(1-n^{2}\right)}}$$

Using n=1.4585 for fused silica, we obtained $n_{\mathrm{eff}}=1.3059$ for p=0.33. We then deduced the longitudinal sound speed *c* from the Brillouin peak positions $\nu_{B}$ in the SBA-15 MSNs using Equation (Equation 3). We obtained c≃1900, 2220, and 2530 m/s for T=550, 700, and 800 °C. These speeds corresponded to longitudinal waves propagating perpendicular to the *c* axis of the hexagonal structure because the elongated NPs lied flat on the surface. Therefore, they represented about 79% of the longitudinal sound speed in the silica walls $c_{L}$ (see Figure 7 (right)), leading to $c_{L}≃2400$, 2810, and 3200 m/s. These values were lower than for the MCM-41 MSNs. We assigned this difference to the microporous structure of SBA-15. Indeed, contrary to MCM-41, SBA-15 is a dual-porosity system with micropores perpendicular to the hexagonal channels [33,34]. We expected these micropores to weaken the silica walls, resulting in lower sound speeds. The volume of the micropores obtained from the N_2_ physisorption isotherms was 10, 4.8, and 2.4% of the total porous volume for T=550, 700, and 800 °C, respectively. This showed that the micropores closed when increasing the calcination temperature. In addition, contrary to MCM-41, the hexagonal structure of SBA-15 was not damaged at 800 °C. These were the reasons why *c* increased between 700 and 800 °C. However, it did not reach the value obtained for the non-porous silica NPs or the MCM-41 NPs calcined at 700 °C.

We did not observe peaks in the optical phonon range for the MSNs using Brillouin or low-frequency Raman scattering. However, as discussed in the Introduction, peaks at larger frequencies (>300 GHZ) were reported in a previous work for mesoporous titania [9]. To attempt a qualitative comparison, we performed the phonon bands’ calculations using an isotropic approximation of anatase TiO_2_ ($c_{L}=8326$ and $c_{T}=3863\;m/s$) and the porosity and pore diameter reported in that study, assuming the same 2D hexagonal mesoporous lattice structure. Indeed, the structure should be close to that of SBA-15 because the parameters of the synthesis were similar. We obtained a quadrupolar-like Raman active mode at νa≃3800 m/s (E_2g_) and a breathing-like one at νa≃11,600 m/s (A_1g_). Other Raman active modes existed in between, as in Table 3. The experimental values (2650 and 4760 m/s) fell within this range. A definitive assignment was of course not possible at this point. In particular, it is first necessary to confirm that the mesoporous structure is ordered and hexagonal so that the approach presented in this work can be applied.

## 5. Conclusions

We presented a method to evaluate the stiffness of the walls in an ordered mesoporous structure from Brillouin light scattering measurements and finite element calculations of the band structure of the corresponding phononic crystal. MCM-41 MSNs were synthesized, and their structure was carefully characterized. The samples calcined at T=550 and 700 °C had an ordered periodic mesoporous structure suitable for comparison with phononic crystal calculations. The sound speed in the silica walls deduced from their Brillouin spectra and calculations increased with the calcination temperature up to a value similar to the one we measured for non-porous silica NPs. This encouraging result supported our approach to use a phononic crystal description to obtain the stiffness of the silica walls. Smaller sound speeds were obtained for the MCM-41 MSNs calcined at 800 °C and the SBA-15 MSNs. They were assigned to variations of the mesoporous structure due to the collapse of the MCM-41 structure above 700 °C and to micropores in the dual-porosity system of the SBA-15. This showed that the mesoporous and wall structures can play a significant role in the sub-THz vibrational dynamics. No signature of the optical phonon of the ordered mesoporous structure was observed in this work. Previous results obtained for mesoporous titania hinted that such modes may be observable in other ordered mesoporous materials. The present work did not demonstrate unambiguously that acoustic waves in ordered mesoporous silica can be described with a continuum elasticity phononic crystals approach. However, this system can be varied in many ways, which can help confirm this possibility. For example, pore expanders can be used to obtain larger spatial periodicities and thicker walls for which the validity of continuum elasticity would be less debatable. Other mesoporous structures also exist, which would be potentially more interesting for phononic applications such as cubic ones, which are suitable to obtain a complete band gap [11]. Additionally, the pores may be filled with a different inorganic material. Hopefully, this work will help open the door to such applications in the future.

## Figures and Tables

**Figure 1 nanomaterials-13-02078-f001:**
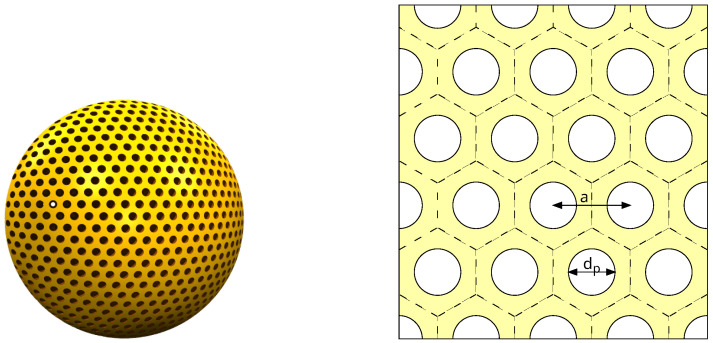
**Left**: drawing of a typical MCM-41 MSN. **Right**: cross-section of the 2D hexagonal lattice of cylindrical pores of the MCM-41 mesoporous structure. The hexagonal cells are delimited by dashed lines. *a* is the distance between the centers of the pores and $d_{p}$ the pore diameter.

**Figure 2 nanomaterials-13-02078-f002:**
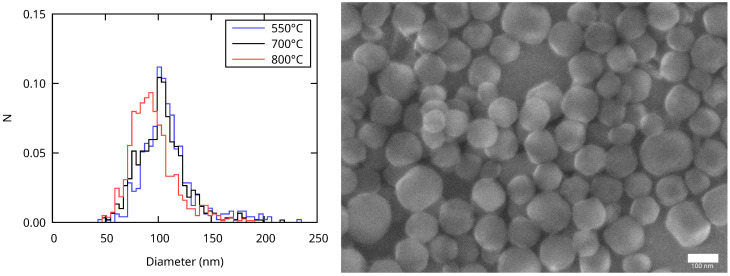
Size distributions of the MCM-41 MSNs calcined at different temperatures (**left**) and representative SEM image of the sample calcined at 700 °C (**right**; scale bar: 100 nm).

**Figure 3 nanomaterials-13-02078-f003:**
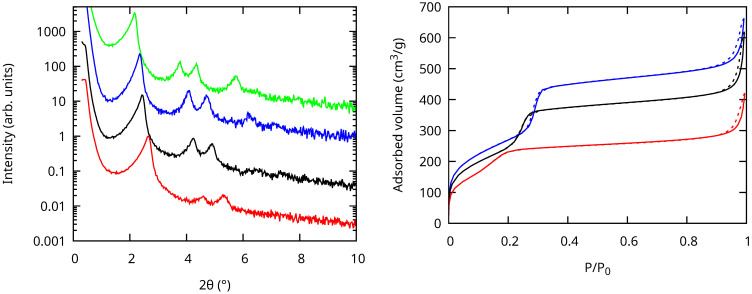
XRD patterns (**left**) and N_2_ physisorption isotherms (**right**) of the MCM-41 MSNs calcined at T=800 °C (red), 700 °C (black), 550 °C (blue), and before calcination (green, XRD only) from bottom to top. The XRD patterns are vertically shifted for clarity. The full and dashed curves are the adsorption and desorption branches of the isotherm, respectively.

**Figure 4 nanomaterials-13-02078-f004:**
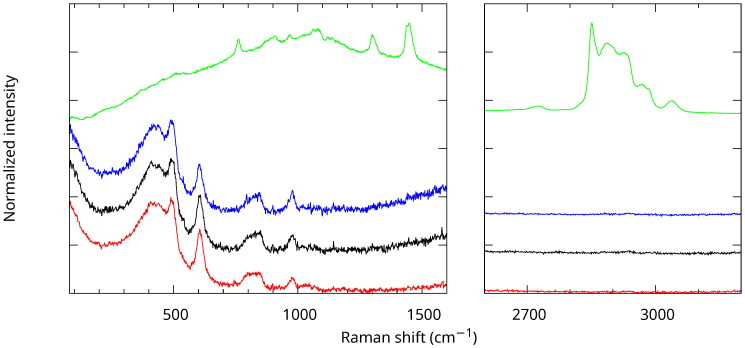
Raman spectra of the MCM-41 MSNs calcined at T=800 °C (red), 700 °C (black), 550 °C (blue), and before calcination (green) from bottom to top. The intensities of the calcined samples were normalized, and the spectra are vertically shifted for clarity.

**Figure 5 nanomaterials-13-02078-f005:**
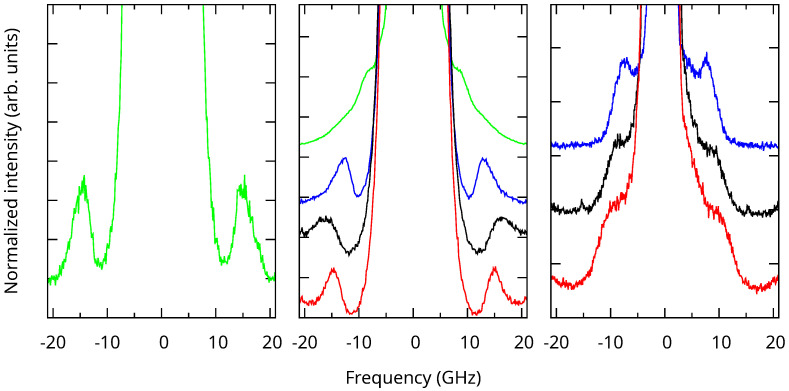
Brillouin spectra of the uncalcined SiO_2_ NPs (**left**), the MCM-41 (**center**), and SBA-15 (**right**) MSNs calcined at 800 °C (red), 700 °C (black), 550 °C (blue), and before calcination (green, MCM-41 only) from bottom to top. The intensities were normalized, and the spectra are vertically shifted for clarity.

**Figure 6 nanomaterials-13-02078-f006:**
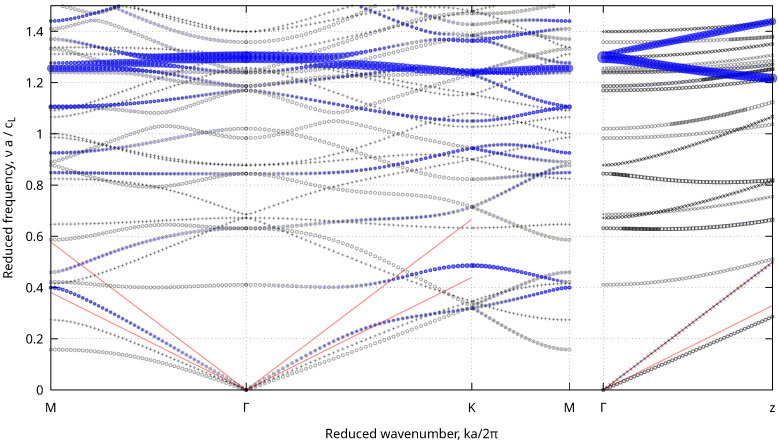
Phonons bands in reduced units for p=0.35. Γ, M, and K are the usual points of the hexagonal Brillouin zone. The right plot is for phonons propagating along the directions of the pores (*z*). The straight red lines starting at Γ have slopes corresponding to $c_{L}$ and $c_{T}$. The blue points show the vibrations, which change the surface area of the silica walls’ cross-sections. The radius of the blue circles is proportional to the surface change.

**Figure 7 nanomaterials-13-02078-f007:**
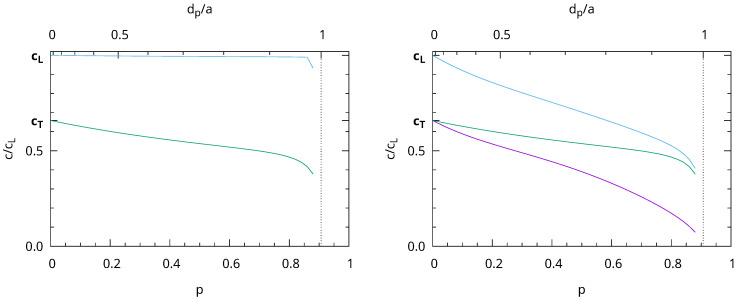
Variation of the sound speeds as a function of porosity for waves propagating along the *z* direction (**left**) and in the xy plane (**right**).

**Figure 8 nanomaterials-13-02078-f008:**
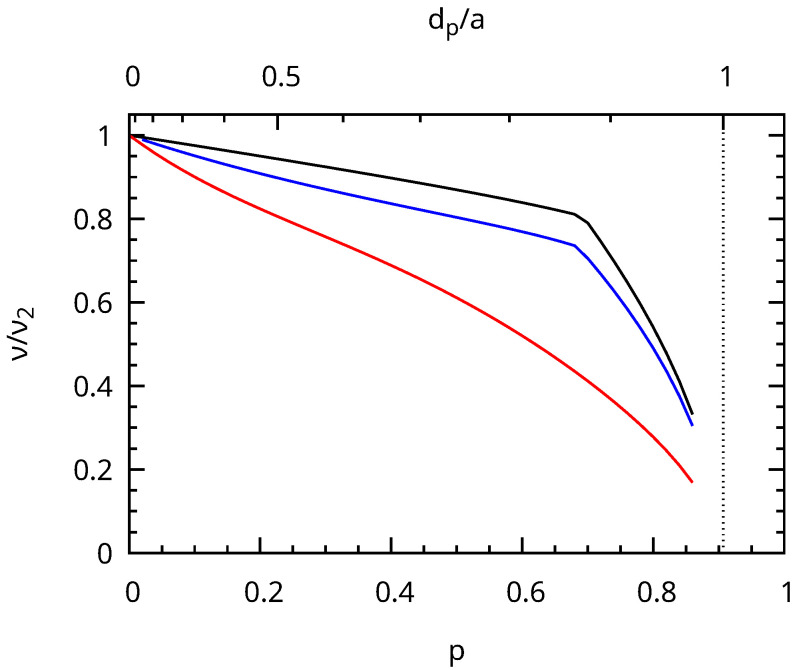
Variation of quadrupolar-like eigenfrequencies of a spherical NP normalized to the frequency of the quadrupolar mode (spheroidal, ℓ=2) of an isotropic sphere without porosity. The E_1g_ (red), E_2g_ (blue), and A_1g_ (black) branches are plotted from bottom to top.

**Figure 9 nanomaterials-13-02078-f009:**
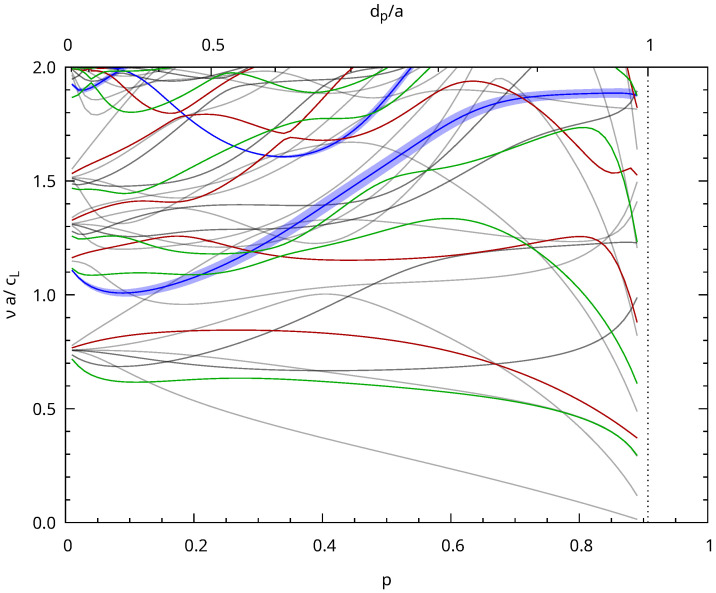
Variation of the reduced frequency of the optical phonons at the center of the Brillouin zone as a function of porosity. The blue curves are for A_1g_ phonons. Their thicknesses are proportional to the variation of the surface area of the silica walls during oscillation. E_1g_ and E_2g_ phonon frequencies are plotted in red and green, respectively. The curves in gray are for Raman inactive phonons.

**Table 1 nanomaterials-13-02078-t001:** Average diameter ($\bar{d}$), lattice constant (*a*), pore diameter ($d_{p}$), wall thickness ($e=a-d_{p}$), $\frac{d_{p}}{a}$, porosity (*p*), and Brillouin peak position as a function of calcination temperature (*T*) for the MCM-41 MSNs.

*T* (°C)	$\bar{d}$ (nm)	*a* (nm)	$d_{p}$ (nm)	*e* (nm)	$d_{p}/a$	*p* (%)	Brillouin (GHz)
550	110	4.3	2.6	1.7	0.60	33	13
700	103	4.2	2.4	1.8	0.57	30	16
800	95	3.8	2.1	1.7	0.55	28	15

**Table 2 nanomaterials-13-02078-t002:** Lattice constant (*a*), pore diameter ($d_{p}$), wall thickness ($e=a-d_{p}$), $\frac{d_{p}}{a}$, porosity (*p*), and Brillouin peak position as a function of the calcination temperature (*T*) for the SBA-15 MSNs.

*T* (°C)	*a* (nm)	$d_{p}$ (nm)	*e* (nm)	$d_{p}/a$	*p* (%)	Brillouin (GHz)
550	9.784	5.88	3.904	0.60	33.8	7.5
700	9.149	5.59	3.559	0.61	33.9	8.8
800	8.781	5.23	3.551	0.60	32.2	10

**Table 3 nanomaterials-13-02078-t003:** Deformation of the pore surface of the first Raman active vibrations at Γ for mesoporous silica with p=0.35. Dotted lines show the surface of the pores at rest and the hexagonal cells. Only one of the degenerate modes is illustrated for the E modes (degeneracy 2).

i.r.	E_2g_	E_1g_	E_1g_	E_2g_	A_1g_
$\frac{\nu a}{c_{L}}$	0.61	0.81	1.13	1.17	1.28
	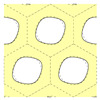	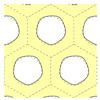	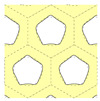	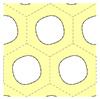	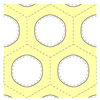

## Data Availability

The datasets used and/or analyzed during the current study are available from the corresponding author upon reasonable request.

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
