# Peer review of "Sub-THz Vibrational Dynamics in Ordered Mesoporous Silica Nanoparticles"

_nanomaterials, 2023, doi:10.3390/nano13142078_

Round 1

Reviewer 1 Report

In this paper, the authors performed Brillouin scattering measurements on several mesoporous silica nanoparticles and evaluated the sound velocity of silica walls in ordered mesoporous structures using finite element calculations. The authors evaluated the basic properties of mesoporous silica nanoparticles and discussed them based on comparisons with some previous studies. I agree with this paper that the experimental results are reliable and that it is valuable for those in the field of phononics. 

However, I see the broad Brillouin scattering peaks (particularly in SBA-15) in Fig. 5, so I suspect the uncertainty of evaluated Brillouin peaks (shown in Tables 1 and 2) and the estimated sound velocity. Before the publication, I consider that the authors must show the fitting results  etc. in the supplementary material and show the deviation of the evaluated value.

Author Response

We first would like to thank the reviewer for the constructive comments and suggestions which helped us significantly improve our manuscript.

In this paper, the authors performed Brillouin scattering measurements on several mesoporous silica nanoparticles and evaluated the sound velocity of silica walls in ordered mesoporous structures using finite element calculations. The authors evaluated the basic properties of mesoporous silica nanoparticles and discussed them based on comparisons with some previous studies. I agree with this paper that the experimental results are reliable and that it is valuable for those in the field of phononics.

Thank you for this positive assessment.

However, I see the broad Brillouin scattering peaks (particularly in SBA-15) in Fig. 5, so I suspect the uncertainty of evaluated Brillouin peaks (shown in Tables 1 and 2) and the estimated sound velocity. Before the publication, I consider that the authors must show the fitting results  etc. in the supplementary material and show the deviation of the evaluated value.

We agree with the reviewer that this part of the manuscript should have been more explicit in our original submission. The fitting procedure is now presented as supplementary material. The obtained peak frequencies are quite reliable (asymptotic standard error lower than a few %). A short paragraph was added at the end of section 3.3 to point to the new supplementary material.

Reviewer 2 Report

The experimental and theoretical research activity reported in the manuscript is well structured, interesting and well described. However, a limited series of critical issues prevented the publication of the manuscript in its current state.

Specifically, the following points are highlighted:

1 - The introduction, results and conclusion sections fail to clearly highlight the novelty level of the proposed work and its implications in the field of phonon propagation in nanostructured materials and beyond.

2 - The criteria that led to the choice of the experimental parameters adopted in the experimental study are not clearly highlighted, such as for example the three post-processing temperatures of 550, 700 and 800 °C.

3 - For greater clarity in reading all the figures, it is suggested to add the legends and possibly divide a graph into several graphs with fewer curves.

4 - A comment on the deformation of the pore surface shown in Table 3 is required for clarity.

5 - It would be appropriate to provide, with justification, the level of confidence regarding the isotropic continuum elasticity approximation carried out during the calculation of the phonon bands in the case of the material under study, ie mesaporous silica nanoparticles.

Check for typos and some sentences could be a little more fluent.

Author Response

We first would like to thank the reviewer for the constructive comments and suggestions which helped us significantly improve our manuscript.

The experimental and theoretical research activity reported in the manuscript is well structured, interesting and well described.

Thank you for this positive assessment.

However, a limited series of critical issues prevented the publication of the manuscript in its current state.

Specifically, the following points are highlighted:

1 - The introduction, results and conclusion sections fail to clearly highlight the novelty level of the proposed work and its implications in the field of phonon propagation in nanostructured materials and beyond.

The last paragraph of the introduction and the beginning of the conclusion have been rewritten to emphasize that the goal of the present work is to propose a method to determine the elastic parameters inside the walls of the mesoporous structure. To the best of our knowledge, no such measurements are available elsewhere. It is meaningful because it allows to compare the structure of the material the walls are made of between samples prepared with different conditions. In addition, using ordered mesoporous structures as phononic materials is an interesting possibility which was already presented in the original submission.

2 - The criteria that led to the choice of the experimental parameters adopted in the experimental study are not clearly highlighted, such as for example the three post-processing temperatures of 550, 700 and 800 °C.

This information was indeed lacking in the original manuscript. The temperatures were chosen so that the silica walls are as close as possible to bulk silica. The temperatures are therefore as large as possible without damaging the mesoporous structure. This is now written at the end of the Introduction section.

3 - For greater clarity in reading all the figures, it is suggested to add the legends and possibly divide a graph into several graphs with fewer curves.

We failed to see what to improve. The captions fully describe all the curves in the different figures. Figure 6 may seem complex but it is the standard way to present phonon bands for different propagation directions. The figures with the measurements (XRD, N2 physisorption, Raman, Brillouin) use the same color code for the calcination temperature so that they are easier to understand.

4 - A comment on the deformation of the pore surface shown in Table 3 is required for clarity.

Thank you for this comment. The text associated to this table was indeed too short in the original submission. We expanded it while keeping it relatively short because the manuscript does not present any new experimental evidence of such optical phonons.

5 - It would be appropriate to provide, with justification, the level of confidence regarding the isotropic continuum elasticity approximation carried out during the calculation of the phonon bands in the case of the material under study, ie mesaporous silica nanoparticles.

The main issue regarding this approximation is the thinness of the silica walls. As explained in the text, the isotropic continuum elasticity approximation has been successfully used in the past for much smaller nanostructures. This is possible as long as the calculations involve acoustic phonons whose wavelength is significantly larger than the interatomic distance. The lowest frequency modes are therefore often relevant even for very small sizes (~1nm).

Comments on the Quality of English Language

Check for typos and some sentences could be a little more fluent.

We have double-checked the text of the manuscript and corrected a few errors.

Round 2

Reviewer 1 Report

In this paper, the authors performed Brillouin scattering measurements on several mesoporous silica nanoparticles to evaluate the sound velocity of silica walls. In the last review round, I commented that it was necessary to show the fitting results in the supplementary materials to show the deviation of the evaluation values. The authors clearly show it in their revised manuscript. Therefore, I consider that it is suitable for the publishing as is.